A new golden species of Diasporus (Anura: Eleutherodactylidae) from southwestern Colombia, with evaluation of the phylogenetic significance of morphological characters in Diasporus

Ospina Sarria Jhon Jairo jhon.sarria@fundacioncalima.org 1
Velásquez Trujillo David Andrés 1
Castaño Saavedra Christian Oswaldo 1
Castillo Luis Fernando 2
Bolívar-García Wilmar 1 3
1 Calima, Fundación Para la Investigación de la Biodiversidad y Conservación en el Trópico , Cali , Valle del Cauca , Colombia
2 Calidris, Asociación Para el Estudio y Conservación de las Aves Acuáticas en Colombia , Cali , Valle del Cauca , Colombia
3 Universidad del Valle, Departamento de Biología, Grupo de Investigación en Ecología Animal , Cali , Valle del Cauca , Colombia
Meegaskumbura Madhava
Electronic publication date: 2022 Feb 8
Publication date: 2022
Volume: 10
Electronic Location ID: e12765
Received 2021 Aug 25; Accepted 2021 Dec 17
Copyright: ©2022 Ospina Sarria et al.
Copyright year: 2022
Copyright holder: Ospina Sarria et al.
License: This is an open access article distributed under the terms of the Creative Commons Attribution License, which permits unrestricted use, distribution, reproduction and adaptation in any medium and for any purpose provided that it is properly attributed. For attribution, the original author(s), title, publication source (PeerJ) and either DOI or URL of the article must be cited.
License URL: https://creativecommons.org/licenses/by/4.0/

Keywords: Amphibia, Anchicayá, Diasporus diastema species group, Evolution, Phylogeny, Synapomorphy, Taxonomy, Terrarana

Funding: The Rufford Small Grants Foundation 27658-1 35562-2 Idea Wild Fundación Univalle (Acuerdo de Regalías No. 65 del 12 de Marzo del 2018) Calidris, Asociación para el Estudio y Conservación de las Aves Acuáticas en Colombia and Empresa de Energía del Pacífico (EPSA) Corporación Autónoma Regional del Valle del Cauca (CVC) Universidad del Valle This study was supported by the Rufford Small Grants Foundation (grant no: 27658-1, 35562-2). Idea Wild provided us with an equipment grant to conduct this study successfully. Colombian funding was provided by the Fundación Univalle (Acuerdo de Regalías No. 65 del 12 de Marzo del 2018). The expedition to Bajo Anchicayá was funded by contract EP-C0-302-2011 signed by Calidris, Asociación para el Estudio y Conservación de las Aves Acuáticas en Colombia and Empresa de Energía del Pacífico (EPSA). The expedition to San Cipriano was funded by cooperation agreement 166-2017 signed by Corporación Autónoma Regional del Valle del Cauca (CVC) and Universidad del Valle. The funders had no role in study design, data collection and analysis, decision to publish, or preparation of the manuscript.

==============================
A new species of Diasporus is described from the lowlands of southwestern Colombia. The new species exhibits a yellowish coloration in life, a character that it shares with other three species in the genus—Diasporus citrinobapheus, D. gularis, and D. tigrillo. The new species differs from all other congeners in having two chrome orange spots (=glandlike protrusions) on sacral region, smooth ventral skin, basal webbing between the toes, and distal papillae at tips of disc covers on fingers II–IV and toes II–IV. Further, the new species differs from all congeners by an uncorrected p-distance of > 5.56% of the 16S rRNA gene fragment examined. In addition to the new species described herein, we demonstrated that the possession of a yellowish coloration in life optimizes unambiguously as a synapomorphy of a clade within Diasporus, which may be recognized as the Diasporus diastema species group. We also discussed the phylogenetic significance of two morphological characters previously considered of systematic value in Diasporus, the occurrence of oval palmar tubercles (undivided) and longitudinal folds (of the vocal sacs) on the throat. On this basis, we demonstrated that these characters appear to be symplesiomorphies rather than synapomorphies of Diasporus. Regarding pointed disc covers (ungual flap) present in some species of Diasporus, we show that this character conflates various characters, involving variation in pad shape, dorsal outline of the disc (ungual flap), and dependence between discs of different digits. Finally, considering that phenotypic data are a valuable source of evidence in testing phylogenetic hypotheses of terraranan frogs, we encourage future research to incorporate phenotypic evidence into phylogenetic studies involved in the genus Diasporus.

Introduction

The terraranan genus Diasporus (formerly Eleutherodactylus diastema species group sensu Savage, 1997; Lynch & Duellman, 1997; Lynch, 2001) is presently composed of 16 species, of which three exhibit a yellowish coloration in life—D. citrinobapheus Hertz et al., 2012, D. gularis (Boulenger, 1898), and D. tigrillo (Savage, 1997). In 2011, faunal explorations at the montane forest of the Río Anchicayá, western slopes of the Cordillera Occidental in the Departamento del Valle del Cauca, Colombia, led us to the discovery of a remarkable golden specimen of genus Diasporus. Morphological examination of this specimen revealed a number of significant morphological differences within the genus (i.e., two chrome orange spots on sacral region, smooth ventral skin, basal webbing between the toes, and disc covers of fingers II–IV and toes II–IV with minute papillae) suggesting that it could be considered a new species. Later, the examination of preserved specimens of Diasporus gularis deposited in the amphibian collection at the Instituto de Ciencias Naturales, Universidad Nacional de Colombia, revealed an additional specimen of this golden species collected by John D. Lynch in the 1990s. More recently, in 2018, field surveys yielded one more specimen. All these three specimens come from localities in the vicinity to the Estación Agroforestal Bajo Calima in the Departamento del Valle del Cauca, Colombia, which is the type locality of another two species of Diasporus—D. quidditus (Lynch, 2001) and D. tinker (Lynch, 2001).

Since the last taxonomic review of Diasporus distributed in northern South America conducted by Lynch (2001), no attempt at a morphological revision has been carried out and no new species have been described from this area. A different trend occurs in Middle America where the proliferation of works dealing with Diasporus has significantly improved our understanding of the systematics and taxonomy of this genus (e.g., Chaves et al., 2009; Hertz et al., 2012; Batista et al., 2016; García-Rodríguez, Arias & Chaves, 2016; Arias et al., 2019). Indeed, in Middle America seven species were described in the last nine years (Frost, 2021). But while it is true that this body of knowledge has been favorable for the genus Diasporus, there is still a need to increase the taxon sampling within the genus. In such endeavor, it is necessary to incorporate species distributed in northern South America into future phylogenetic studies involved in overhauling the taxonomy of this genus—Diasporus anthrax (Lynch, 2001), D. gularis, D. quidditus, and D. tinker. Moreover, given that there is a degree of taxonomic uncertainty among some Diasporus species (e.g., D. diastema Cope, 1875, D. hylaeformis Cope, 1875, D. quidditus; see Batista et al., 2016; Arias et al., 2019), it is necessary to carry out a taxonomic revision for Diasporus through combined analyses of phenotypic and molecular evidence.

In this paper we describe a new golden species of Diasporus from southwestern Colombia previously confused with D. gularis. In addition, we provide molecular data for the new species and Diasporus gularis and discussed the phylogenetic significance of two morphological characters previously considered of systematic value in Diasporus, the occurrence of oval palmar tubercles (undivided) and longitudinal folds (of the vocal sacs) on the throat (Cochran & Goin, 1970; Lynch & Duellman, 1997; Lynch, 2001). In particular, our goal with the analysis of phylogenetic significance was to evaluate whether the occurrence of these character states optimize as synapomorphies for Diasporus. Finally, we encourage future research to incorporate phenotypic evidence into the phylogenetic studies involving in the genus Diasporus.

Materials & Methods

Molecular phylogenetic Analyses

We conducted phylogenetic analyses based on DNA sequences of two mitochondrial genes (16S, 1,270 bp and cytochrome oxidase subunit I - COI, 658 bp), and one nuclear gene (recombinase activation 1 - RAG-1, 584 bp). Recombinase activation 1 (RAG-1) gene nucleotide sequence was obtained from GenBank. Total genomic DNA was extracted from ethanol-preserved muscle tissue using the DNeasy Tissue Kit Qiagen according to the manufacturer’s protocol. PCR amplification was carried out in 30-µl reactions containing 12.5-µl GoTaq Green master mix 2X (Promega, USA), 0.25-µl of each forward and reverse gene primers (10 mM), 1-µl of extracted DNA, and 16-µl of grad water. The mitochondrial genes 16S and COI were amplified using the set of primers 16 sar-L/16 sbr-H (Palumbi et al., 1991) and dgLCO-1490/dgHCO-2198 (Meyer, Geller & Paulay, 2005), respectively. The PCR protocol consisted of an initial denaturation step at 95 °C (5 min), 32 cycles consisting of denaturation at 95 °C (30 s), annealing at 50 °C and 48 °C (30 s) for 16S and cytochrome oxidase subunit I, respectively, and extension at 72 °C (40 s) followed by a final extension step at 72 °C (10 min). PCR-products were cleaned using exonuclease plus 1 U of Alkaline Phosphatase per reaction. Purified PCR products were bidirectionally sequenced using a generic analyzer (Macrogen Inc., South Korea). Chromatograms obtained were processed using Sequencher v 5.0.1 (Gene Codes, Ann Arbor, MI, USA). Complete sequences were edited with the software Geneious pro 5.5.7 (Kearse et al., 2012).

For the phylogenetic analyses we included representative sequences of all known and candidate species of Diasporus for which there are available sequences in GenBank (In total, 88 individuals were included in the analysis). Contaminated or misidentified sequences were detected and excluded from our analyses: Diasporus darienensis Batista et al. (2016) (GenBank accession numbers: KT186631 and KT186585) and Diasporus aff. diastema (GenBank accession numbers: KT186633 and KT186588). The sampling outside Diasporus was designed to test hypotheses of monophyly of the genus and character-states polarity of morphological characters. In total, we included sequences of 31 species representing all genera within the family Eleutherodactylidae (Adelophryne Hoogmoed & Lescure, 1984, Eleutherodactylus Duméril & Bibron, 1841, and Phyzelaphryne Heyer, 1977) according to the phylogenetic hypotheses of Hedges, Duellman & Heinicke (2008), Pyron & Wiens (2011), Padial, Grant & Frost (2014), Jetz & Pyron (2018). Ischnocnema lactea (Miranda-Ribeiro, 1923), a species of Brachycephalidae, was used for rooting the trees. Most DNA sequences were obtained from GenBank, except sequences of the new species and Diasporus gularis (Sequences provided for review in Data S1). GenBank accession numbers for the new species, Diasporus gularis, and previously available sequences are provided in the Data S2.

Sequences were aligned using the online server of MAFFT software v7 (http://mafft.cbrc.jp/alignment/server/large.html) (Katoh & Toh, 2008; Katoh, Rozewicki & Yamada, 2019) under the strategy E-INS-i and default parameters for gap opening and extension. Phylogenetic analyses were performed using Maximum-likelihood and Parsimony inference methods. The maximum-likelihood analysis was performed with IQ-TREE v1.6.12 (Nguyen et al., 2015). ModelFinder was used in IQ-TREE to calculate the best partition schemes and best-fit models of substitution for the data sets (Kalyaanamoorthy et al., 2017). The best partition scheme included four subsets (see Table 1). The maximum likelihood analysis (ML) included 1,000 ultrafast bootstrap (UFBoot) replicates (Hoang et al., 2018), with a nearest neighbor interchange search (-bnni) to reduce the risk of overestimating branch support. The parsimony analysis was performed in TNT version 1.5 (Goloboff, Farris & Nixon, 2008; Goloboff & Catalano, 2016), using the New Technology Search algorithms. Gaps were considered as a fifth state and transformation events were equally weighted. In addition, we performed a parsimony analysis considering gaps as missing data for comparisons with the maximum likelihood analysis. Parsimony jackknife absolute frequencies were calculated using the New Technology Search algorithms (Goloboff, 1999; Nixon, 1999), as well as requesting 10 hits with driven searches, for a total of 1,000 replicates. We favoured parsimony (gaps treated as a fifth state) as the optimality criterion because it uses the number of transformations as an optimality criterion to select among heuristic solutions, and propose less costly and easier hypotheses to falsify (Goloboff, 2003; Kluge, 2005; Kluge & Grant, 2006; Farris, 2008; Grant & Kluge, 2009). The resulting trees were visualized through the iTOL phylogenetic tree viewer (Letunic & Bork, 2007) and edited using Inkscape software (XQuartz 2.8.1, http://www.inkscape.org/).

Table 1 Best partition scheme and best-fit models selected by ModelFinder for the IQ-TREE analysis.

Subset	Data blocks	Model	
1	Non coding mitochondrial sequences	TIM2+F+I+G4	
2	Coding mitochondrial sequences 1st, 2nd	TN+F+I+G4	
3	Coding mitochondrial sequences 3rd	TIM+F+G4	
4	Coding nuclear sequences 1st, 2nd, 3rd	TVM+F+G4	

Uncorrected p-distances of the 16S gene sequences were calculated using the software PAUP v.4.0 (Swofford, 2002) for a dataset with all sequences having the same length and no missing data (575 bp including gaps) and containing only sequences from taxa closely-related to the new species. These sequences were aligned using the software MAFFT under the strategy G-INS-i with default parameters for gap opening and extension. Sequences are available in the Dryad Digital Repository (https://datadryad.org/stash/share/knYwt-h_OJOWi3kh_GlH5BPCIUTEqRydAEuQDVNFkgk).

Morphology

For morphological analysis, we followed definitions and terminology provided by Lynch & Duellman (1997) and Lynch (2001) and the standardized format for definitions (diagnoses) provided by Duellman & Lehr (2009) for terraranan frogs. Sex was determined by examination of external secondary sexual character (vocal slits) and by direct inspection of gonads. Fingers and toes are numbered pre- to post-axially from I to IV and I to V, respectively. To estimate lengths of toes III and V, we adpressed both toes against Toe IV, and for lengths of fingers I and II, we appressed those fingers against each other. Measurements were taken to 0.1 mm with dial or digital calipers. Abbreviations are as follows: SVL = snout–vent length, HL = head length, IOD = interorbital distance. Institutional abbreviations are: CPZ-UV Colección de Prácticas Zoológicas Universidad del Valle, Cali, Colombia; ICN (Instituto de Ciencias Naturales, Museo de Historia Natural, Universidad Nacional de Colombia, Bogotá), KU (Biodiversity Institute, University of Kansas), and MHUA-A (Museo de Herpetología, Universidad de Antioquia, Medellín, Colombia). All specimens examined are listed in Data S3.

Ancestral state reconstruction

Among the morphological characters shared by most species of the genus Diasporus, it has been suggested that the occurrence of oval palmar tubercles (undivided) and longitudinal folds (of the vocal sacs) on the throat can have systematic value since they are uncommon in the superfamily Brachycephaloidea (Savage, 1997; Lynch & Duellman, 1997; Lynch, 2001). On this basis, we defined two characters to describe relevant variation in palmar tubercles and external vocal sacs. In addition to these characters, we also scored the presence of a yellowish coloration in life for all species in our data set. The definitions of the morphological characters and their states are as follows:

Character 0. Palmar tubercle shape: 0 oval, 1 bifid (see fig 3, Lynch, 2001).

Character 1. External vocal sacs: 0 not forming folds, 1 forming longitudinal folds (see fig 1, Lynch, 2001).

Character 2. Coloration in life: 0 brown to reddish, 1 yellowish (see figs 6–9, Lynch, 2001).

The sources of evidence used to score these characters include examination of museum specimens, publications, and field observations. Details on taxonomic distribution for each character and literature sources are listed as Data S4. Ancestral character states were reconstructed on the topology obtained with parsimony (gaps treated as a fifth state) using YBYRÁ (Machado, 2015) and TNT v.1.5 (Goloboff & Catalano, 2016) to identify and plot synapomorphies. YBYRÁ generates color-coded boxes to indicate if a derived state occurs only in the clade in question (non-homoplastic) or also occurs in other clades (homoplastic) and if it is shared by all terminals of the clade (unique) or is subsequently transformed into one.

Our research was conducted under the authorization of the National Authority of Environmental Licences and the Ministry of Environment and Sustainable Development of Colombia (Resolución 1070 del 28 de Agosto de 2015). The electronic version of this article in Portable Document Format (PDF) will represent a published work according to the International Commission on Zoological Nomenclature (ICZN), and hence the new names contained in the electronic version are effectively published under that Code from the electronic edition alone. This published work and the nomenclatural acts it contains have been registered in ZooBank, the online registration system for the ICZN. The ZooBank LSIDs (Life Science Identifiers) can be resolved and the associated information viewed through any standard web browser by appending the LSID to the prefix http://zoobank.org/. The LSID for this publication is: urn:lsid:zoobank.org:pub:149A419D-DF15-424F-BB6E-1D41894F1DB4. The online version of this work is archived and available from the following digital repositories: PeerJ, PubMed Central SCIE and CLOCKSS.

Results

Molecular phylogenetic analyses

The phylogenetic analysis using parsimony as optimality criterion (gaps treated as a fifth state) resulted in 30 most-parsimonious trees of 5,879 steps. One of the trees is shown in Fig. 1. The conflict between these optimal trees involves relationships among conspecific specimens of all species of Diasporus, except in D. amirae Arias et al. (2019), and D. sapo Batista et al. (2016). According to our molecular phylogeny, which agrees with earlier studies (e.g., Hedges, Duellman & Heinicke, 2008; Padial, Grant & Frost, 2014; Jetz & Pyron, 2018), Diasporus is the sister clade of Eleutherodactylus and these in turn form a clade that is sister to a clade formed by the genera Adelophryne and Phyzelaphryne (Fig. 1). The topological differences between the parsimony and the ML analyses involve clades with jackknife values below 50% in the parsimony analyses and ML ultrafast bootstrap support value of 76%, associated with the position of Diasporus vocator (Taylor, 1955) and the clade composed of D. aff. hylaeformis plus two terminals—Diasporus sp. (MHCH 1678 and SMF 97652) (Figs. S1–S2). Considering gaps as a fifth state, the two terminals of Diasporus sp. (MHCH 1678, SMF 97652) were recovered as D. vocator, with <50% jackknife support (Fig. 1); a similar position is recovered when gaps are treated as missing data (Fig. S1). On the other hand, we recovered these two specimens as another distinct lineage in ML (Fig. S2, 96% ultrafast bootstrap support value), as obtained by Arias et al. (2019).

Figure 1 Phylogenetic relationships of Diasporus and outgroups recovered in one of the most parsimonious trees of 5,879 steps from a parsimony analysis in TNT treating gaps as a fifth state.

Black dots indicate nodes that collapse in the strict consensus. Numbers on nodes are Parsimony jackknife absolute frequencies values. Nodes without values indicate value <50% of jackknife value and an asterisk indicates a 100%. Character numbers is given beneath square, with primitive-derived characters states inside square (see text and Data S4 for character definitions). Color-coding: blue = Transformed, homoplastic.

The specimen of the new golden species is recovered as the sister taxon to a clade containing Diasporus gularis, D. aff diastema EPL, D. aff diastema MM, D. tigrillo, D. diastema, and D. citrinobapheus across all the analyses with low support (<50% jackknife support in the parsimony analyses and 56% ultrafast bootstrap support value in ML; Fig. 1; Figs. S1–S2). The uncorrected p-distances showed a relatively high genetic differentiation of the new species compared to other species of Diasporus (5.56–8.72%; Table 2). On the basis of phylogenetic and phenotypic evidence, below we provide a formal description of the new species.

Diasporus lynchisp. nov.	
LSID urn:lsid:zoobank.org:act:6A8D759A-0094-459A-B92B-36D18A1D037D	
LSID urn:lsid:zoobank.org:pub:149A419D-DF15-424F-BB6E-1D41894F1DB4 (Figs. 2A–2B, 3–4)	
Proposed standard Spanish name. Rana Dorada de Anchicayá	
Proposed standard English name. Anchicaya’s Golden Frog	

Holotype.—CPZ-UV 7298 (field no. JJS 065), an adult male obtained by Jhon Jairo Ospina-Sarria on August 4, 2011, at San Marcos, 54 m elevation, on the Río Tatabro (= tributary of Río Anchicayá, 3°41′N, 76°56′W; datum = WGS84), Departamento del Valle del Cauca, Colombia.

Table 2 Percentage of uncorrected p-distances of the 16S gene of Diasporus species most related to D. lynchi sp. nov. (in bold) in the phylogenetic analysis.

Taxon (n)	1	2	3	4	5	6	
1 Diasporus aff diastema MM (1)	0.00						
2 Diasporus citrinobapheus (10)	4.37–5.10	0.00					
3 Diasporus diastema (5)	5.09–6.06	2.16–4.10	0.00				
4 Diasporus gularis (1)	6.28	7.70–8.37	6.98–8.38	0.00			
5 Diasporus tigrillo (5)	5.13	3.38–4.33	3.84–5.04	7.91	0.00		
6 Diasporus lynchi (1)	6.10	5.56–6.53	5.57–6.29	8.72	6.32	0.00	

Figure 2 Living specimens of the Diasporus species known from Colombia.

Diasporus lynchi (A–B; holotype, CPZ-UV 7298 (field no. JJS 065), adult male, snout–vent length [SVL] 19.1 mm; photo: J.J Ospina-Sarria); Diasporus gularis (C; not collected, Pianguita, Valle del Cauca; J.J Ospina-Sarria); Diasporus quidditus (D; holotype, ICN 45173, adult male; photo: J.D. Lynch); Diasporus anthrax (E; MHUAA 09775, San Vicente de Chucuri, Santander; photo: H. Martinez); Diasporus tinker (F; holotype, ICN 45174, adult male; photo: J.D. Lynch).

Figure 3 Dorsal and ventral views of Diasporus lynchi (holotype, CPZ-UV 7298 (field no. JJS 065) in life (A, B) and preservative (C, D). Photos: J.J. Ospina-Sarria.

Paratypes.—Two adult males (ICN 13292 and CPZ-UV 05934). ICN 13292 (field no. JDL 13541) obtained by John D. Lynch at Campamento Agua Bonita, Estación Agroforestal Bajo Calima, 300 m elevation (3°59′N, 76°46′W; datum = WGS84), Departamento del Valle del Cauca, Colombia. CPZ-UV 05934 obtained by Eliana Barona on July 12, 2018, at San Cipriano, 104 m elevation, on the Río Escalerete (=tributary of Río Dagua, 3°49′N, 76°53′W; datum = WGS84), Departamento del Valle del Cauca, Colombia.

Diagnosis.—Diasporus lynchi is diagnosed by the following combination of characters: (1) skin on dorsum and venter smooth; discoidal fold absent; dorsolateral folds absent; (2) tympanic membrane absent; tympanum annulus visible through skin round, its length 33–40% of eye length in two males; supratympanic fold poorly defined; (3) snout subacuminate in dorsal view, truncate in profile; canthus rostralis angular, weakly concave; loreal region slightly concave; (4) upper eyelid bearing two or three small tubercles; narrower than IOD (54.1–60% IOD); cranial crest absent; (5) choanae small, ovoid; partially concealed by palatal shelf of maxillary arch; dentigerous processes of vomers prominent and positioned posterior to level of choanae, triangular in outline, separated medially by a distance equal to the width of the visible dentigerous process, each dentigerous process of vomers bearing four to six teeth; (6) males having vocal slits and large subgular vocal sac (forming longitudinal folds, Figs. 3B, 3D); nuptial pads absent; testes white; (7) finger I shorter than finger II; discs slightly wider than digits, disc on finger I smaller than that of finger II and this in turn smaller than discs on fingers III and IV; disc cover on finger I unornamented; papillae on digits II, III, and IV; triangular pads on fingers; (8) fingers with lateral fringes; palmar tubercle oval (undivided; Fig. 4A); thenar tubercle oval, equal in size to palmar tubercle; supernumerary tubercles low, restricted to the proximal segments of fingers III and IV; subarticular tubercles low, with rounded base and larger than supernumerary tubercles; (9) ulnar tubercles absent; (10) heel and tarsus lacking tubercles and folds; (11) oval inner metatarsal tubercle, its length four times its width; low, conical outer metatarsal tubercle one-fourth size of inner metatarsal tubercle; supernumerary plantar tubercles absent; subarticular present; (12) toes bearing lateral fringes; webbing basal, I 2−- 2+ II 2−- 3+ III 3−- 4+IV 4−- 21/2 V (Fig. 4B); toe III shorter than toe V; toe III reaching midway between penultimate and distal subarticular tubercle of toe IV; toe V extending to distal edge of distal subarticular tubercle of toe IV; discs of toes III–V larger than disc of toe II and this in turn larger than discs on toe I; discs covers on toe I and V unornamented; discs with minute papillae at tips of toes II, III, and IV; triangular pads on toes; (13) in life, dorsal ground color yellow with dark markings, anterior and posterior surfaces of thighs chrome orange. Two chrome orange spots (=glandlike protrusions) on sacral region (Fig. 3A). Canthal, interorbital, and postocular stripes are poorly defined. Limbs with darks marking and disc covers blackish gray (Fig. 2A). Ventral surfaces of body almost transparent with scattered iridophores, ventral surfaces of hind limbs chrome orange, and vocal sac yellow with diminutive black spots (Fig. 3B). The iris is golden-bronze with a reddish-brown horizontal streak; (14) SVL in three adult males 19.1, 19.5, and 19.7 mm.

Figure 4 Ventral view of hand showing lateral fringes on fingers, papillae on digits II, III, and IV, and plantar view of foot showing basal webbing in Diasporus lynchi (A, CPZ-UV 7298, B, CPZ-UV 05934; photos: J.J. Ospina-Sarria). Scale bar = 1 mm.

Comparisons with congeners.—Diasporus lynchi is distinctive in the genus by having a yellowish coloration in life, two chrome orange spots (=glandlike protrusions) on sacral region, smooth ventral skin, basal webbing between the toes, and disc covers of fingers II–IV and toes II–IV with minute papillae (Fig. 2). By having a yellowish coloration in life, D. lynchi requires comparison with D. citrinobapheus, D. gularis, and D. tigrillo. From all these species, D. lynchi differs by having two chrome orange spots (=glandlike protrusions) on sacral region (Fig. 3A). In D. lynchi, basal webbing occurs between toes whereas tips of disc covers on fingers II–IV and toes II–IV have papillae (both character states absent in D. citrinobapheus and D. tigrillo, (Hertz et al., 2012; Savage, 1997). D. lynchi and D. gularis share the occurrence of basal webbing between toes and small papillae on Toes II–IV. However, D. lynchi has smooth ventral skin (areolate in D. gularis, Lynch & Duellman, 1997) and choanae partially concealed by palatal shelf of maxillary arch (choanae not concealed by palatal shelf of maxillary arch in D. gularis Lynch & Duellman, 1997).

Diasporus lynchi can be easily distinguished from the remaining species of the genus Diasporus by having basal webbing between toes (absent in D. amirae, D. anthrax, D. darienensis Batista et al., 2016, D. diastema, D. igneus Batista, Ponce & Hertz, 2012, D. majeensis Batista et al., 2016, D. pequeno, D. sapo, and D. ventrimaculatus Chaves et al., 2009; Arias et al., 2019, Chaves et al., 2009, Batista, Ponce & Hertz, 2012, Batista et al., 2016, Lynch, 2001, Lynch & Duellman, 1997), dentigerous processes of vomers prominent (absent in D. vocator and D. hylaeformis Savage, 2002), and ventral surfaces of body almost transparent with scattered iridophores (venter brown with cream flecks in D. quidditus and gray to dark brown with white blotches in D. tinker; Lynch, 2001).

Description of the holotype.—An adult male with head as broad as body; head width 38.7% of SVL; HL 34% of SVL; snout long, subacuminate in dorsal view and truncate in profile; eye–nostril distance 64% of diameter of eye; nostrils protuberant, directed laterally. Canthus rostralis angular, canthal stripe dark; loreal region slightly concave lacking tubercles; lips no flarep; internarial region depressed; top of head flat; upper eyelid bearing two low tubercles, its width 60.0% of IOD; supratympanic poorly defined, tympanic membrane not evident, tympanic annulus present, one postrictal tubercle posteroventral to tympanic annulus. Choanae small, nearly round, partially concealed by palatal shelf; dentigerous processes of vomers prominent and posteromedian to choanae, triangular in outline, each process bearing four teeth; tongue much longer than broad, its posterior border not notched, posterior third not adherent to floor of mouth; paired vocal slits present, longitudinal, lateral to base of tongue; external vocal sac forming longitudinal folds (Figs. 3B, 3D).

Skin on dorsum smooth, skin on belly, throat, and ventral surfaces of the thighs smooth; discoidal fold absent; cloacal sheath short; no tubercles in cloacal region. Ulnar tubercles absent. Thenar tubercle oval, equal in size to palmar tubercle; supernumerary palmar tubercles at base of fingers III and IV; subarticular tubercles round, flattened and larger than supernumerary palmar tubercles; fingers having lateral fringes; relative lengths of fingers I <II <IV <III, papillae on discs on fingers II, III, and IV; nuptial pads absent. Hind limbs moderately robust; when hind limbs flexed perpendicular to axis of body, heels do not overlap; tibia length 33% of SVL; foot length 28.2% of SVL; heel and tarsus lacking tubercles and folds; oval inner metatarsal tubercle, its length four times its width; low, conical outer metatarsal tubercle one-fourth size of inner metatarsal tubercle; toes bearing lateral fringes; basal webbing, I 2−- 2+ II 2−- 3+ III 3−- 4+IV 4−- 21/2 V; discs with minute papillae at tips of toes II, III, and IV; relative lengths of toes I < II < III < V < IV; fifth toe much longer than third, toe V extending to distal edge of distal subarticular tubercle of toe IV, toe III reaching midway between penultimate and distal subarticular tubercle of toe IV. Supernumerary plantar tubercles absent, subarticular tubercles rounded and flattened.

In life, dorsal ground color bright yellow with black or brown reticulations, two chrome orange spots (=glandlike protrusions) on sacral region, and black canthal, interorbital, and postocular stripes. Limbs with darks marking and upper surfaces of discs blackish gray. Hind limbs chrome orange. Vocal sac pale yellow with small black spots and belly almost transparent with scattered iridophores. The palmar and plantar surfaces are dark gray with the digit pads pale gray. The iris is golden-bronze with a reddish-brown horizontal streak. In preservative, the bright yellow and orange colors have faded to a dull yellow, but the black reticulations on dorsum, blackish gray on upper surfaces of discs, and dark gray on palmar and plantar surfaces remain (Figs. 3C–3D).

Measurements of holotype (mm).—SVL 19.1, tibia length 6.4, foot length 5.4, HL 6.5, head width 7.4, IOD 3.0, internarial distance 2.5, width of upper eyelid 1.5, diameter of eye 2.5, eye–nostril distance 1.6, diameter of tympanum annulus 1.

Distribution and ecology.—The holotype was found vocalizing in leaf litter along a forested stream in a primary forest at night. The calls sound like a “whistle”. The paratypes were also found along stream in primary but thinned forest. Both paratypes were found at distances no greater than 2 m from the stream. Thus, the species seems to be associated with streams in primary forests. As in other species of Diasporus (e.g., D. gularis and D. tinker), D. lynchi vocalizes from concealed sites (e.g., dried leaf); therefore, a considerable effort is required to detect each individual. The species occurs at low elevations in the humid tropical regime (54–300 m elevation) in localities in the vicinity to the Estación Agroforestal Bajo Calima, Departamento de Valle del Cauca, Colombia (Fig. 5), which is the locality type of Diasporus quidditus and D. tinker.

Figure 5 Map of southwestern Colombia (inset) showing localities of Diasporus lynchi: San Marcos (star), San Cipriano (Black circle), and Bajo Calima (white circle).

Lines indicate boundaries of department within which the new species is known to occur.

Etymology.—The specific name is an noun in the genitive case and is a patronym for John D. Lynch, who first found the species during his explorations of the Bajo Calima, and in recognition of his many contributions to understanding the taxonomy and systematics of the world’s most diverse family-group of amphibians (superfamily Brachycephaloidea = Terraranae).

Ancestral state reconstruction

In the context of the phylogenetic hypothesis favoured herein (Fig. 1), we found that none of the character states related to the variation of palmar tubercles and external vocal sac optimize as a synapomorphy for Diasporus. Conversely, we found that the occurrence of a yellowish coloration in life optimizes as synapomorphy of a clade within Diasporus including D. aff. diastema EPL, D. aff. diastema MM, D. citrinobapheus, D. diastema, D. gularis, D. lynchi, and D. tigrillo.

Discussion

Diasporus was recovered as monophyletic and sister clade to Eleutherodactylus, in agreement with earlier reported phylogenies (e.g., Hedges, Duellman & Heinicke, 2008; Pyron & Wiens, 2011; Padial, Grant & Frost, 2014; Jetz & Pyron, 2018). Within Diasporus, most of the recovered clades are poorly supported in the parsimony and ML analyses (Fig. 1, Figs. S1 –S2), suggesting that our knowledge about the phylogenetic relationships of Diasporus is still incomplete (Figs. S1 –S2). Based on our results and those of previous analyses (e.g., Batista et al., 2016; Arias et al., 2019), a more intensive character and taxon sampling is critical to rigorously test the phylogenetic relationships within this genus.

With the description of the golden Diasporus lynchi, seventeen species are currently recognized in Diasporus. Diasporus gularis and D. lynchi were recovered within a clade in which most species exhibit a yellowish coloration in life (D. citrinobapheus, D. gularis, D. lynchi, and D. tigrillo) with <50% jackknife support (Fig. 1). Among the species comprising this clade, the only exception to the yellowish coloration pattern occurs in Diasporus diastema, which according to the original description exhibits a dark brown coloration (Cope, 1875). Regarding this matter, it is important to point out that the description of Diasporus diastema by Cope (1875) was based on specimens collected by John Bransford during the Panama Canal Survey of 1874, and therefore, it is not clear whether the dark brown coloration recorded by Cope (1875) for Diasporus diastema is the coloration in life or in preservative. More recently Batista et al. (2016) reported that Diasporus diastema is composed of three divergent lineages, eastern Panamanian lowlands (EPL), Majé mountain range (MM) and central Panama (CP). Remarkably, the yellowish coloration in life appears in two of those lineages (EPL and MM, see fig 15: Batista et al., 2016). Considering that the presence of a yellowish coloration in life optimizes unambiguously as a morphological synapomorphy of a clade within Diasporus comprising of D. aff. diastema EPL, D. aff. diastema MM, D. citrinobapheus, D. diastema, D. gularis, D. lynchi, and D. tigrillo, we propose to recognize this clade as the Diasporus diastema species group.

Phylogenetic significance of oval palmar tubercles (undivided) and longitudinal folds (of the vocal sacs) on the throat in Diasporus

Our results demonstrated that none of the character states related to the variation of palmar tubercles and external vocal sac optimize as a synapomorphy for Diasporus. Oval palmar tubercles (undivided) also appear in Eleutherodactylus and Adelophryne (see Data S4). In addition to these findings, the occurrence of oval palmar tubercles (undivided) also has been reported in Phyzelaphryne miriamae (Hoogmoed & Lescure, 1984). A similar situation was noted for the occurrence of longitudinal folds (of the vocal sacs) on the throat. Current evidence shows that longitudinal folds (of the vocal sacs) on the throat also occur in some species of Eleutherodactylus from Cuba and Hispaniola (e.g., Lynch & Duellman, 1997; Lynch, 2001; Hedges, Duellman & Heinicke, 2008). Likewise, the occurrence of longitudinal folds (of the vocal sacs) on the throat has been reported in Adelophryne adiastola (Hoogmoed & Lescure, 1984: fig 6). Based on these results, the occurrence of oval palmar tubercles (undivided) and longitudinal folds (of the vocal sacs) on the throat could be judged to be tentatively a symplesiomorphy rather than synapomorphy of Diasporus. Future works increasing taxon sampling outside Eleutherodactylidae will further elucidate whether the occurrences of these characters represent synapomorphies of Eleutherodactylidae.

Comments on the pointed disc covers (ungual flap) in Diasporus

Although Savage (1997) and Lynch (2001) proposed delimiting the morphological variation of the disc covers in four states (i.e., palmate, spadate, lanceolate, and papillate), such a system of codification involves a logical dependency among several independent characters. Particularly, the system of codification proposed by Savage (1997) and Lynch (2001) entails dependence between the morphologies of ungual flap and disc pads, and at the same time, dependence between discs of different digits. Current empirical evidence suggests that ungual flap and disc pads morphologies vary independently, and these in turn vary independently between fingers (Ospina-Sarria & Grant, 2021). Following Ospina-Sarria & Grant’s proposal, future works investigating the variation in morphology of disc covers must explicitly delimit variation in terms of the morphology of the disc pads and the morphology of ungual flap across different fingers (I, II, and III–IV) as independent characters. To our knowledge, pointed disc covers also occur in Adelophryne and Phyzelaphryne (Hoogmoed & Lescure, 1984), but until transformational series are objectively delimited, it is not possible to test the phylogenetic significance of morphological variation of digital discs in Diasporus.

Conclusions

With the description of Diasporus lynchi, 17 species compose the genus Diasporus. Phylogenetic analyses based on molecular evidence indicate that D. lynchi is the sister species of a clade containing D. gularis, D. aff. diastema EPL, D. aff. diastema MM, D. tigrillo, D. diastema, and D. citrinobapheus. Furthermore, we found that the possession of a yellowish coloration in life optimizes unambiguously as a morphological synapomorphy for this clade. Although species of genus Diasporus are characterized by having oval palmar tubercles (undivided) and longitudinal folds (of the vocal sacs) on the throat, none of them optimizes unambiguously as synapomorphies of this genus. Another morphological character that has been suggested to characterize the species of Diasporus is the pointed disc covers (ungual flap), however, there is evidence revealing that multiple characters have been conflated under this character.

Frogs of the genus Diasporus, together with the genus Pristimantis, are among the terraranan genera with the greatest taxonomy uncertainty for many of their taxa. This is because, as mentioned earlier, no attempt at morphological revision has been conducted recently. In the specific case of Diasporus, this situation has led to a lack of confidence in the taxonomic identification of numerous DNA sequences deposited in GenBank (e.g., Diasporus diastema, D. hylaeformis, D. quidditus, D. tinker). To address this situation, it is necessary a taxonomic revision incorporating morphological evidence. Besides, and considering that many species of Diasporus are sympatric and syntopic, we recommend being cautious when assigning species identification based on specimens collected near or at the type locality. For example, the Estación Agroforestal Bajo Calima in the Departamento de Valle del Cauca, Colombia is the type locality of Diasporus quidditus and D. tinker, but D. gularis and D. lynchi can also be found. That is, at the Estación Agroforestal Bajo Calima is possible to find four of the five species of the genus Diasporus known from Colombia, except for D. anthrax that is distributed in the Valle of the Magdalena River (Frost, 2021).

Finally, as has been pointed out in the literature, phenotypic data are a valuable source of evidence in testing phylogenetic hypotheses of terraranan frogs. Therefore, we hope our results encourage further research using phenotypic characters into phylogenetic and taxonomic studies involving the genus Diasporus.

Supplemental Information

Supplemental Information 1 DNA sequences

Click here for additional data file.

Supplemental Information 2 GenBank accession numbers for the sequences employed in the phylogenetic analyses

Species in bold are those for which we provide original sequences. COI, cytochrome oxidase subunit I; RAG1, recombinase activation.

Click here for additional data file.

Supplemental Information 3 Specimens Examined

Click here for additional data file.

Supplemental Information 4 Transformation series and literature sources using in the parsimony ancestral state reconstruction. See text for character definitions. Inapplicable characters are scored using a dash

Click here for additional data file.

Supplemental Information 5 Phylogenetic relationships of Diasporus and outgroups recovered in one of the most parsimonious trees of 5,418 steps from a parsimony analysis in TNT treating gaps as missing data

Black dots indicate nodes that collapse in the strict consensus. Numbers on nodes are Parsimony jackknife absolute frequencies values. Nodes without values indicate value <50% of jackknife value and an asterisk indicates a 100%.

Click here for additional data file.

Supplemental Information 6 Phylogenetic relationships of Diasporus and outgroups recovered in the maximum likelihood analysis with IQ-TREE (log-likelihood of consensus tree: -35559.2884)

Numbers around nodes are the ultrafast bootstrap support values. An asterisk (*) indicates 100% ultrafast bootstrap support.

Click here for additional data file.

We are grateful to J.D. Lynch (ICN), R. Brown, R. Glor, L. Welton, W.E Duellman (KU), and Neftali Camacho (Natural History Museum of Los Angeles County) for loans of specimens, workspace, and the many other courtesies provided on numerous occasions. Also, we thank J.D. Lynch and J.M. Daza (MHUA-A) for providing photos of Diasporus anthrax, D. quidditus, and D. tinker. We especially thank Eliana Barona, who provided data of the paratype from San Cipriano. L. Barrientos generously provided the sequences used for the new species and D. gularis. For their assistance and hospitality, we are grateful to the local communities from Anchicayá and San Cipriano, Valle del Cauca, Colombia.

Additional Information and Declarations

Competing Interests

Author Contributions

Animal Ethics

Field Study Permissions

DNA Deposition

Data Availability

New Species Registration

The authors declare there are no competing interests.

Jhon Jairo Ospina Sarria and David Andrés Velásquez Trujillo conceived and designed the experiments, performed the experiments, analyzed the data, prepared figures and/or tables, authored or reviewed drafts of the paper, and approved the final draft.

Christian Oswaldo Castaño Saavedra conceived and designed the experiments, performed the experiments, prepared figures and/or tables, authored or reviewed drafts of the paper, and approved the final draft.

Luis Fernando Castillo and Wilmar Bolívar-García conceived and designed the experiments, prepared figures and/or tables, authored or reviewed drafts of the paper, and approved the final draft.

The following information was supplied relating to ethical approvals (i.e., approving body and any reference numbers):

National Authority of Environmental Licences and the Ministry of Environment and Sustainable Development of Colombia.

The following information was supplied relating to field study approvals (i.e., approving body and any reference numbers):

National Authority of Environmental Licences and the Ministry of Environment and Sustainable Development of Colombia.

The following information was supplied regarding the deposition of DNA sequences:

The sequences are available at GenBank: MZ871499, MZ871500, and MZ881958, and in the Supplemental File.

The following information was supplied regarding data availability:

The sequences used in the uncorrected p-distances analysis of the 16S mitochondrial rRNA fragment gene are available at Dryad Digital Repository: Ospina-Sarria, Jhon Jairo et al. (2022), Data from: A new golden species of Diasporus (Anura: Eleutherodactylidae) from southwestern Colombia, with evaluation of the phylogenetic significance of morphological characters in Diasporus, Dryad, Dataset, https://doi.org/10.5061/dryad.j3tx95xdv.

The following information was supplied regarding the registration of a newly described species:

Publication LSID: urn:lsid:zoobank.org:pub:149A419D-DF15-424F-BB6E-1D41894F1DB4

Diasporus lynchi sp. nov.

LSID urn:lsid:zoobank.org:act:6A8D759A-0094-459A-B92B-36D18A1D037D

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
