# Peer review of "A new golden species of Diasporus (Anura: Eleutherodactylidae) from southwestern Colombia, with evaluation of the phylogenetic significance of morphological characters in Diasporus"

_PeerJ, doi:10.7717/peerj.12765_

## Round 0.1 · original submission · Major Revisions

I have now received three reviewer reports for your study. Their recommendations range from Minor to Major Revisions, and they all agree that the species described is divergent enough to be valid. However, they require you to provide more information and clarity in distinguishing the species. These are the major points that need improvements related to your paper.

1. Greater clarity in explaining the phylogenetic analyses and presenting the ML based phylogeny.

2. Improvements in morphology, especially related to Hand and Foot, together with the addition of a figure.

3. If possible, adding a comparative bioacoustics-related analysis, or at least describing the call of the new species.

4. I also encourage you to carry out a species delimitation analysis to distinguish the new species using a 2-4 species delimitation methods. These are very straightforward methods that you can execute quickly.

However, please consider all the issues raised by the reviewers and provide a point-by-point rebuttal with your revision.

Reviewer 1 ·

Basic reporting

I think the article is pertinent, since the contribution of this new species is valid and is a contribution to the diversity of this group. In general, the article is well written. But the phylogenetic analyses are poor, they require a detailed review, in particular the alignments on which they are made. I also believe that it is very important that the results of Maximum likelihood are explicitly put in the document and that the results of the phylogenetic part are not only based on the results of the Parsimony analyzes, since all previous works for this group report results. Based on ML and to be clearly comparable, it is necessary that they be exposed, which is not clearly the case in the article.

Experimental design

The methods are not complete, data on the protocol used for the RAG1 nuclear gene are lacking.

Information is also lacking to clarify the number of individuals included in the analysis and the length of the sequences analyzed.

Why the authors performed alignments with default values of the programs if the nature of each of the genes is different, they should not be the same costs in the alignments for protein-coding genes as the ribosome-associated genes (16S)
the cost of insertion and extension of gaps must be different, this will affect the alignments and therefore the results of the phylogenetic inferences

Validity of the findings

The taxonomic results are valid, but the phylogenetic approach is poor and requires that the alignments and therefore the analyses and results of the phylogenetic inferences be reviewed.

Additional comments

line 95 Add to the line a more integrative approach, to include molecular and morphological data is not enough, also is required behavior, acoustics, and maybe biochemical signaling to understand this group of frogs.

Line 233, The authors should also include the most recent phylogenetic hypotheses about the positions of this group of frogs and which are based on massive data from the position of Diasporus tales Hutter 2017, Barrientos 2021. This may give further support to the relationships with respect to their group brother (Eleutherodactylus) which are quite clear.

Line 235 and fig1 Authors should show ML results or summarize them in the parsimony tree that authors show in figure 1 or show both trees

·

Basic reporting

no comment

Experimental design

no comment

Validity of the findings

The authors gives relevant nformation for the understanding of the genus, and only minor revision is required to be published.

The authors must add figures of a hand and a foot to see details of characters (fringes, and webbing) that they have stated are important for the recognition of the species.
Other comments are in the revised manuscript

Additional comments

Other comments are embedded into the revised manuscript

Reviewer 3 ·

Basic reporting

The text is well-written and good to understand. However, it would certainly be good to check the text once again. There are some minor grammar errors. However, the authors use clear, unambiguous, and technically correct language. Literature references are sufficient.
The Introduction should include all the relevant work on Diasporus. I noticed that e.g., the works of Arias et al. 2019 and Chavez et al. 2009 are not mentioned in the introduction. The work of Garcia-Rodriguez et al. 2016 (Neotropical Biodiversity Volume 2, 2016 - Issue 1) is not mentioned at all.
A weak point in this description is the lack of any bioacoustics data or analysis. Diasporus calls are easy to record and the analysis is cheap and easy. Call descriptions have been proven a powerful tool for species identification, especially in species groups that lack good morphological characteristics. Describing a new species of Diasporus without presenting any bioacoustic data, call description, and a comparison with its congeners is not appropriate any more.
The general structure of the text is good.

Experimental design

I generally welcome this study and I agree with the authors that there is a lack of work in this genus from South America, while there has been considerable progress in Central America. I also don´t doubt that the species presented by the authors is new to science.
I, however, have some issues with the approach the authors use to delimit the new species.
First, I think that sample size of five appears a little low. Diasporus species are usually abundant where they occur. Especially coloration may be very variable in Diasporus species. This is one of the main characteristics (two chrome orange spots (=glandlike structures) on sacral region) the authors use to distinguish the new species and I see the small sample size as a weak point here.
An even bigger weak point in this description is the lack of any bioacoustics data or analysis. Diasporus calls are easy to record and the analysis is cheap and easy. Call descriptions have been proven a powerful tool for species identification, especially in species groups that lack good morphological characteristics. Describing a new species of Diasporus without presenting any bioacoustic data, call description, and a comparison with its congeners is not appropriate any more.

Validity of the findings

I think describing the species as new is justified. The genetic distance to the closest relatives is large enough and there are some morphological characters that support that idea. However, including a bioacoustics analysis would provide an additional line of evidence, especially when comparing with calls of Diasporus gularis and the other yellowish species.
Further, I would be careful with putting too much emphasis on the idea of a D. diastema clade that includes only bright yellow species. First, not all species are bright yellow and second the clade is not supported (<50% jackknife support) in the phylogenetic analysis.

Additional comments

I have made some other, mostly minor, comments directly in the pdf.

Annotated reviews are not available for download in order to protect the identity of reviewers who chose to remain anonymous.

---

## Round 0.2 · accepted · Accept

The authors have addressed the comments by the reviewers to a reasonable degree, with a justification of approach regarding the type of tree that should be presented in the paper. The main focus of the paper is clear in that there is no ambiguity in the new species justification. Hence, the paper now meets the editorial requirements needed to be accepted. I congratulate the authors on the description of a new Diasporus species, and wish them the best with their future research.

·

Basic reporting

See below

Experimental design

See below

Validity of the findings

See below

Additional comments

I have read the rebuttal letter and the manuscript, and I do not have additional comments on it.

Reviewer 3 ·

Basic reporting

Dear authors,

I am happy with most of your reply and welcome, the additional work you have spent to improve your manuscript.
I am still in a way disappointed that there will be no information on the male advertisement call of this species. I have never doubted that you discovered a new species and I think and have always thought your two lines of evidence are enough for species delimitation. I also don´t think that call data are more important than the other lines of evidence and I never said that. However, for a complete description of a new species that emits a sound, reporting on that sound is very important, as it is a very important aspect in the life of the animal and a significant information to identify it in the future. Imagine you would describe a frog on molecular and bioacoustic evidence only, but would not give a description on how it looks like. Yes, of cause you can do your species description without bioacoustics and I intentionally did not ask for an extensive bioacoustic analysis (although for many Diasporus species there is excellent bioacoustic material available), but asked for any bioacoustic data like a simple recording you made with a cell phone, likewise the photos you took. Simple parameters like the call length or the frequency range would have been interesting for researchers working on that genus. To keep it short, I accept that you don´t have call data available at the moment and hope anyone will describe Diasporus lynchi´s call someday in the future.

Comments in the PDF
Abstract: Since the new species is not the only yellow species in the genus, this sentence is kind of redundant.
Reply: We respectfully disagree with this suggestion. We think that it is not redundant; on the contrary, it is very informative in the context of the results, discussion and conclusions of our work.
Reply to the reply: No, you got me wrong here! My comment was unclear, sorry my fault. My comment was a pure linguistic thing. You wrote “The new species, along with Diasporus citrinobapheus, D. gularis, and D. tigrillo are the only species in this genus known to exhibit a yellowish coloration in life.” First, it must read “the new species is” not "are"! And, what I meant with my comment is basically the word “only”. It would be like saying “Bananas, along with cherries, apples, pears, and peaches, are the only sweet fruits produced by plants”. You know what I mean? I would just reword the sentence saying something like: “The new species exhibits a yellowish coloration in life, a character that it shares with other species in the genus like Diasporus citrinobapheus, D. gularis, and D. tigrillo.” Or something in that way.

Apart from that I am completely fine with the manuscript.

Thanks for choosing me as a reviewer!

Experimental design

no comment

Validity of the findings

no comment